# Sex differences in febrile children with respiratory symptoms attending European emergency departments: An observational multicenter study

Chantal D. Tan[1], Soufiane el Ouasghiri[1], Ulrich von Both[2,3], Enitan D. Carrol[4,5], Marieke Emonts[6,7,8], Michiel van der Flier[9,10,11], Ronald de Groot[9], Jethro Herberg[12], Benno Kohlmaier[13], Michael Levin[12], Emma Lim[6,14], Ian K. Maconochie[15], Federico Martinon-Torres[16], Ruud G. Nijman[12], Marko Pokorn[17], Irene Rivero-Calle[16], Maria Tsolia[18], Clementien L. Vermont[19], Werner Zenz[13], Dace Zavadska[20], Henriette A. Moll[1], Joany M. Zachariasse[1]*, On behalf of PERFORM consortium (Personalised Risk assessment in febrile children to optimise Real-life Management across the European Union)[¶]

1 Erasmus MC-Sophia Children's Hospital, Department of General Paediatrics, Rotterdam, The Netherlands, 2 Division of Paediatric Infectious Diseases, Dr. von Hauner Children's Hospital, University Hospital, Ludwig-Maximilians-University Munich, Munchen, Germany, 3 German Centre for Infection Research, DZIF, Partner site Munich, Munchen, Germany, 4 University of Liverpool, Institute of Infection, Veterinary and Ecological Sciences Liverpool, United Kingdom, 5 Alder Hey Children's NHS Foundation Trust, Liverpool, United Kingdom, 6 Great North Children's Hospital, Paediatric Immunology, Infectious Diseases & Allergy, Newcastle upon Tyne Hospitals NHS Foundation Trust, Newcastle upon Tyne, United Kingdom, 7 Translational and Clinical Research Institute, Newcastle University, Newcastle upon Tyne, United Kingdom, 8 NIHR Newcastle Biomedical Research Centre based at Newcastle upon Tyne Hospitals NHS Trust and Newcastle University, Newcastle upon Tyne, United Kingdom, 9 Section of Paediatric Infectious Diseases, Laboratory of Medical Immunology, Radboud Center for Infectious Diseases, Radboud Institute for Molecular Life Sciences, RadboudUMC, Nijmegen, The Netherlands, 10 Paediatric Infectious Diseases and Immunology, Amalia Children's Hospital, RadboudUMC, Nijmegen, The Netherlands, 11 Paediatric Infectious Diseases and Immunology, Wilhelmina Children's Hospital, University Medical Centre Utrecht, Utrecht, The Netherlands, 12 Section of Paediatric Infectious Diseases, Imperial College, London, United Kingdom, 13 Medical University of Graz, Department of General Paediatrics, Graz, Austria, 14 Population Health Sciences Institute, Department of Medicine, Newcastle University, Newcastle upon Tyne, United Kingdom, 15 Paediatric Emergency Medicine, Imperial College Healthcare Trust NHS, London, United Kingdom, 16 Hospital Clínico Universitario de Santiago de Compostela. Genetics, Vaccines, Infections and Paediatrics Research group (GENVIP), Santiago de Compostela, Spain, 17 University Medical Centre Ljubljana, Department of Infectious Diseases and Faculty of Medicine, University of Ljubljana, Ljubljana, Slovenia, 18 National and Kapodistrian University of Athens, Second Department of Paediatrics, P. and A. Kyriakou Children's Hospital, Athens, Greece, 19 Erasmus MC-Sophia Children's Hospital, Department of Paediatric Infectious diseases and Immunology, Rotterdam, The Netherlands, 20 Rīgas Stradiņa universitāte, Department of Paediatrics, Children clinical university hospital, Riga, Latvia

¶ Membership of the PERFORM consortium is provided in the S1 File.
* j.zachariasse@erasmusmc.nl

**Data Availability Statement:** The minimal anonymized dataset has been uploaded to a public repository: https://doi.org/10.14469/hpc/10644.

## Abstract

### Objective

To assess sex differences in presentation and management of febrile children with respiratory symptoms attending European Emergency Departments.

**Funding:** This project has received funding from the European Union's Horizon 2020 research and innovation programme to ML under grant agreement No. 668303 and No. 848196. The Research was supported by the National Institute for Health Research Biomedical Research Centres at Imperial College London to RGN (CL-2018-21-007), Newcastle Hospitals NHS Foundation Trust and Newcastle University to ME. The views expressed are those of the author(s) and not necessarily those of the NHS, the NIHR or the Department of Health. For the remaining authors no sources of funding were declared. The funders had no role in study design, data collection and analysis, decision to publish, or preparation of the manuscript.

**Competing interests:** The authors have declared that no competing interests exist.

### Design and setting

An observational study in twelve Emergency Departments in eight European countries.

### Patients

Previously healthy children aged 0–<18 years with fever ($\geq$ 38˚C) at the Emergency Department or in the consecutive three days before Emergency Department visit and respiratory symptoms were included.

### Main outcome measures

The main outcomes were patient characteristics and management defined as diagnostic tests, treatment and admission. Descriptive statistics were used for patient characteristics and management stratified by sex. Multivariable logistic regression analyses were performed for the association between sex and management with adjustment for age, disease severity and Emergency Department. Additionally, subgroup analyses were performed in children with upper and lower respiratory tract infections and in children below five years.

### Results

We included 19,781 febrile children with respiratory symptoms. The majority were boys (54%), aged 1–5 years (58%) and triaged as low urgent (67%). Girls presented less frequently with tachypnea (15% vs 16%, p = 0.002) and increased work of breathing (8% vs 12%, p<0.001) compared with boys. Girls received less inhalation medication than boys (aOR 0.82, 95% CI 0.74–0.90), but received antibiotic treatment more frequently than boys (aOR 1.09, 95% CI 1.02–1.15), which is associated with a higher prevalence of urinary tract infections. Amongst children with a lower respiratory tract infection and children below five years girls received less inhalation medication than boys (aOR 0.77, 95% CI 0.66–0.89; aOR 0.80, 95% CI 0.72–0.90).

### Conclusions

Sex differences concerning presentation and management are present in previously healthy febrile children with respiratory symptoms presenting to the Emergency Department. Future research should focus on whether these differences are related to clinicians' attitudes, differences in clinical symptoms at the time of presentation and disease severity.

## Introduction

Sociodemographic characteristics such as sex have been shown to influence health care delivery and outcome [1]. Identifying these differences is crucial for not only optimizing health care outcomes but also minimizing existing inequity in health care [2]. Sex is defined as a biological classification of living things as male or female according to their reproductive organs and functions [3]. Several studies have been conducted regarding sex differences in adults presenting to the Emergency Department (ED), finding clinically relevant differences. For example, a study in adults has demonstrated that men and women have a different presentation of myocardial infarction [4]. Another study has shown that sex-specific protocols for diagnosis,

management, and counseling can influence patient outcomes in sport-related injury in emergency medicine [5].

Although sex differences in adults attending the ED have received increasing attention, research on sex differences in children remains scarce, especially in emergency medicine [6]. The first study to assess the role of sex in pediatric emergency medicine using a multicenter and international cohort found evidence of sex-specific differences regarding management of children after adjustment for age, triage urgency and clinical presentation. This study was conducted in a large heterogeneous population of children using pooled data and they found that boys present more frequently to the ED and receive inhalation medication more often when presenting with respiratory symptoms [7]. Additionally, it has been shown that sex hormones play a role in developmental and physiological differences in the lungs of children both before and during the neonatal period and boys have a higher risk of developing asthma than girls during childhood [8, 9]. This is also reflected in the higher incidence rate of respiratory syncytial virus bronchiolitis in boys compared with girls [10, 11].

Globally, fever and respiratory tract symptoms are one of the most common symptoms among children presenting at the ED and are responsible for 20–25% of all pediatric emergency visits [12–14]. The aim of this study is to examine sex-specific differences regarding presentation and management in a large cohort of febrile children with respiratory symptoms visiting European EDs. Insight in these sex differences may increase our understanding on the role of sex in pediatric emergency medicine and whether these are based on clinical symptoms or physician's attitude.

## Methods

### Study design

The MOFICHE study (Management and Outcome of Fever In Children in Europe) is a European observational multicenter study assessing management and outcome of febrile children in Europe using routine emergency care data and is embedded in the PERFORM project (Personalised Risk assessment in Febrile illness to Optimise Real-life Management across the European Union) [15].

The study was approved by the ethical committees of all the participating hospitals: Austria (Ethikkommission Medizinische Universität Graz, ID: 28–518 ex 15/16), Germany (Ethikkommission der LMU München, ID: 699–16), Greece (Ethics committee, ID: 9683/18.07. 2016), Latvia (Centrala medicinas etikas komiteja, ID: 14.07.201 6. No. Il 16–07–14), Slovenia (Republic of Slovenia National Medical Ethics Committee, ID: ID: 0120-483/2016-3), Spain (Comité Autonómico de Ética de la Investigación de Galicia, ID: 2016/331), The Netherlands (Commissie Mensgebonden onderzoek, ID: NL58103.091.16), United Kingdom (Ethics Committee, ID: 16/LO/1684, IRAS application no. 209035, Confidentiality advisory group reference: 16/CAG/0136). The need for informed consent was waived by the ethics committee. In all the participating UK settings, an additional opt-out mechanism was in place.

### Study population and setting

We included children aged 0–<18 years attending the ED with fever (≥ 38°C) or a history of fever (fever within 72 hours before ED visit) and respiratory symptoms. We decided to focus on febrile children with respiratory symptoms, since the majority of children had respiratory symptoms (63%) and a previous study regarding sex-specific differences in pediatric emergency care has shown that sex differences were present in children attending the ED with respiratory symptoms [7]. Respiratory symptoms were defined as runny nose, coughing, sore throat or sneezing. Twelve EDs from eight European countries (Austria, Germany, Greece,

Latvia, the Netherlands (n = 3), Spain, Slovenia and the United Kingdom (n = 3)) participated in this study. The participating hospitals were either university or large teaching hospitals. More details are described in a previous publication [16]. We excluded children with missing data on management and children with comorbidity, which was defined as having a chronic underlying condition expected to last at least one year [17].

## Data collection

Data were collected from January 2017 to April 2018 for at least one year to account for seasonal variation. Data were collected as part of routine clinical care at the ED and these were extracted from patient records before being entered into electronic case report forms by the local research teams. Data collected included age, sex, triage urgency, comorbidity (as stated in the ED charts or previous history forms), presenting symptoms, vital signs (tachycardia, tachypnea, hypoxia), diagnostic tests (laboratory test, respiratory test, imaging), treatment (antibiotics, inhalation medication, oxygen therapy), disposition, focus of infection and cause of infection. Data on presenting symptoms were restricted to prespecified complaints including respiratory, gastrointestinal (diarrhea, vomiting) and neurological symptoms (seizures or focal neurological signs or meningeal signs), and children were allowed to have multiple presenting symptoms. Age-specific cut off values for vital signs according to Advanced Pediatric Life Support guidelines were used to define tachypnea, tachycardia and hypoxia [18]. The focus of infection was allocated retrospectively by the research team. We used the following categories: upper respiratory tract, lower respiratory tract, urinary tract and other (gastrointestinal tract, childhood exanthemas/flu-like illness, soft tissue or skin/musculoskeletal, sepsis/meningitis, undifferentiated fever and inflammatory illness). The cause of infection, defined as presumed bacterial (definite bacterial, probable bacterial and bacterial syndrome), unknown bacterial or viral, presumed viral (definite viral, probable viral and viral syndrome) or other (e.g. inflammatory), was determined by the research team using a previously published phenotyping algorithm, which combines clinical data, microbiology results and C-reactive protein (CRP) (S1 Fig) [19, 20].

## Outcome measures

Management was defined as diagnostic tests, treatment and admission. We categorized diagnostic tests into general bloodwork (CRP, white blood cell count (WBC) and procalcitonin (PCT)), respiratory test/culture, blood culture, and chest X-ray. Treatment included antibiotic treatment, inhalation medication (salbutamol, ipratropium, epinephrine, budesonide) and oxygen therapy. Admission was defined as both admission to the ward or the Pediatric Intensive Care Unit.

## Data analysis

First, descriptive statistics were used for children's characteristics and management stratified by sex. Chi-squared tests and Mann-Whitney U tests were used assuming not normally distributed data. Second, multivariable logistic regression analyses were performed to examine the association between management and sex, with boys as reference group. We adjusted for the following covariates: age, triage urgency, ill appearance (classify as ill if ill, irritable or uncomfortable is stated in the chart written by triage nurses or physicians), duration of fever, vital signs (tachycardia, tachypnea, hypoxia), increased work of breathing and participating ED. Increased work of breathing was defined as the presence of any of chest wall retractions, nasal flaring, grunting or apnea. Third, we stratified the analyses for children with 1) the upper respiratory tract and 2) the lower respiratory tract as focus of infection. Subsequently, we

performed a subgroup analysis in children up to five years to reduce the influence of pubertal sex hormones, since the rise of adrenal androgens occurs around the age of six to eight years [21, 22]. Based on our results, we performed an additional sensitivity analysis with exclusion of children with the urinary tract as focus of infection. Adjusted odds ratios (aORs) were calculated and a 95% confidence interval (CI) was given. Multiple imputation using MICE package in R were used for missing data on covariates. Statistical analyses were performed using IBM SPSS Statistics software version 25. A p-value below 0.05 was determined as statistically significant.

## Results

### Description of the study population

We included 19,781 previously healthy febrile children with respiratory symptoms after excluding 18% (4321/24,380) of the children due to comorbidity and subsequently excluding 1% (278/20,059) due to missing data on outcome measures. The majority were boys (54%), in the age group of 1–5 years (58%) and triaged as low urgent (67%). Sixty-four percent had an upper respiratory tract infection, 19% had a lower respiratory tract infection and most infections were of viral cause. Sixteen percent had another focus of infection, which could be due to concomitant infection next to their respiratory infection. Girls had less frequently tachypnea (15% vs 16%, p = 0.002) and increased work of breathing (8% vs 12%, p<0.001) compared with boys, whereas girls had more often tachycardia (28% vs 23%, p<0.001) (Table 1).

Management stratified by sex is shown in Table 2 and the range per ED is shown in S1 Table. Simple diagnostics were performed in 39% in boys and in 40% in girls of which the majority consisted of CRP and WBC. Advanced diagnostic tests were performed in 35% in boys and in 36% in girls of which chest X-rays were most often performed. Antibiotics were prescribed in boys and girls in 31% and 33%, inhalation medication in 13% and 10%, oxygen therapy in 3% and 2%, respectively, and both boys and girls were admitted in 20%.

### Association between sex and management

In the total group of 19,781 children, girls received less inhalation medication compared with boys (aOR 0.82, 95% CI 0.74–0.90), but girls received antibiotic treatment more often compared with boys (aOR 1.08, 95% CI 1.02–1.15) as shown in Fig 1. The unadjusted odds ratios are shown in S2 Table.

In the upper respiratory tract subgroup of 12,554 children no significant associations were found between sex and management as shown in Fig 2. The unadjusted odds ratios are shown in S3 Table.

In the lower respiratory tract subgroup of 3808 children, girls received less inhalation therapy compared with boys (aOR 0.77, 95% CI 0.66–0.89) as shown in Fig 3. The unadjusted odds ratios are shown in S4 Table.

In the 14,967 children below five years of age, girls received less inhalation medication compared with boys (aOR 0.80, 95% CI 0.72–0.90) as shown in S2 Fig and S5 Table.

## Discussion

### Main findings

Sex differences regarding presentation and management are present in a large cohort of febrile children with respiratory symptoms attending European EDs. Girls present less often to the ED with fever and respiratory symptoms, and when they do so they have less frequently tachypnea (15% vs 16%) and increased work of breathing (8% vs 12%) compared with boys.

**Table 1. Patient characteristics stratified by sex (N = 19,781).**

|  | Boys (N = 10,870 54%) | Girls (N = 8911 46%) | Total (N = 19,781) |
|---|---|---|---|
| **Age** (years) |  |  |  |
| <1 | 2037 (19) | 1481 (17) | 3518 (18) |
| 1<5 | 6297 (58) | 5152 (58) | 11,449 (58) |
| 5<12 | 2030 (19) | 1789 (20) | 3819 (19) |
| 12–17 | 506 (4) | 489 (5) | 995 (5) |
| **Triage urgency*** |  |  |  |
| Low urgent | 7236 (67) | 6078 (68) | 13,313 (67) |
| High/intermediate urgent | 3336 (31) | 2556 (29) | 3892 (30) |
| **Duration of fever (days)**~ | 1.5 (0.5–3.0) | 1.5 (0.5–3.0) | 18,600 (94) |
| **Ill appearing** | 1383 (13) | 1144 (13) | 2527 (13) |
| **Vital signs** |  |  |  |
| Tachypnea* | 1770 (16) | 1307 (15) | 3077 (16) |
| Tachycardia* | 2509 (23) | 2510 (28) | 5019 (25) |
| Hypoxia | 272 (3) | 207 (2) | 479 (2) |
| Increased work of breathing* | 1260 (12) | 743 (8) | 2003 (10) |
| **Focus of infection**¥* |  |  |  |
| Upper respiratory tract | 6897 (64) | 5657 (64) | 12,554 (64) |
| Lower respiratory tract | 2218 (20) | 1590 (18) | 3808 (19) |
| Urinary tract | 64 (0.6) | 217 (2.4) | 281 (1.4) |
| Other | 1691 (16) | 1447 (16) | 3138 (16) |
| **Cause of infection*** |  |  |  |
| Presumed bacterial | 2153 (20) | 2040 (23) | 4211 (21) |
| Unknown bacterial or viral | 1438 (13) | 1226 (14) | 2664 (14) |
| Presumed viral | 6758 (62) | 5244 (59) | 12,002 (61) |
| Other | 420 (4) | 336 (4) | 756 (4) |

Absolute numbers and percentages (%) are shown

~ median and interquartile range (IQR) 25–75

*p-value <0.05

¥ Most clinically relevant focus of infection was assigned

Missing data: <3% triage urgency, ill appearing, cause of infection, 6–14% duration of fever, tachycardia, hypoxia, increased work of breathing, 23% tachypnea

Few differences in management between boys and girls exist, but we observed consistently lower proportions of girls receiving inhalation medication after adjustment for patient characteristics and markers of disease severity (aOR 0.82, 95% CI 0.74–0.90). This is in line with a previous study in which girls received less inhalation medication compared with boys in children with respiratory symptoms attending the ED (pooled OR 0.79, 95% CI 0.73 to 0.86) [7]. However, these children had respiratory symptoms in general and did not look at febrile children specifically. The lower proportion of girls receiving inhalation medication could be explained by boys having higher rates of wheezing and bronchial hyperresponsiveness during childhood than girls [23, 24]. However, children with comorbidity including asthma were excluded in our study, so this group does not contribute to the sex-specific differences. In the total group we also found that girls more often received antibiotic treatment (aOR 1.08, 95% CI 1.02–1.15) compared with boys. This might be explained by a higher prevalence of a urinary tract infection in girls than boys (2.4% vs 0.6%). A sensitivity analysis where we excluded children with a urinary tract infection (N = 281) did not show a significant difference in antibiotic prescription in girls and boys (aOR 1.03, 95% CI 0.97–1.10). There was no significant

**Table 2. Management stratified by sex (N = 19,781).**

|  | **Boys (N = 10,870)** | **Girls (N = 8911)** |
|---|---|---|
| **Diagnostics** |  |  |
| CRP | 4204 (39) | 3529 (40) |
| WBC | 4193 (39) | 3526 (40) |
| PCT$^\mu$ | 178 (2) | 100 (1) |
| Respiratory test/culture | 2113 (19) | 1802 (20) |
| Blood culture | 681 (6) | 577 (7) |
| Chest X-ray | 1850 (17) | 1524 (17) |
| **Antibiotic treatment** | 3316 (31) | 2923 (33) |
| **Inhalation medication** | 1399 (13) | 883 (10) |
| **Oxygen therapy** | 282 (3) | 203 (2) |
| **Admission** | 2214 (20) | 1751 (20) |

Absolute numbers and percentages (%) are shown

$^\mu$ PCT was performed in 7 out of 12 ED settings

difference in antibiotic prescription rates between boys and girls in the subgroup analyses performed. In addition, no sex differences were found regarding diagnostic tests, treatment with oxygen therapy and admission. This is in contrast to a previous study where differences were present between girls and boys, with the proportion of children receiving laboratory tests and imaging during ED visits for respiratory problems was higher in girls. However, in this previous study subgroup analyses in children with respiratory symptoms and children with fever were performed separately and they have adjusted for other covariates including age, triage urgency and clinical presentation [7].

## Strengths and limitations

This is the first study to examine sex differences in carefully phenotyped febrile children with respiratory symptoms in a large multicenter cohort in twelve European EDs, which makes the results generalizable to the majority of European countries. Clinical data was extensively

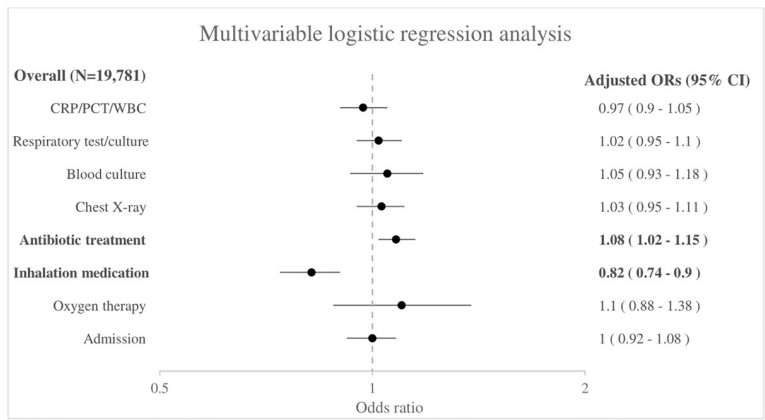

**Fig 1. Association between sex and management for all febrile children presenting with respiratory symptoms (N = 19,781).** Boys as reference group. Adjusted for age, triage urgency, ill appearance, tachypnea, tachycardia, hypoxia, work of breathing, duration of fever, ED.

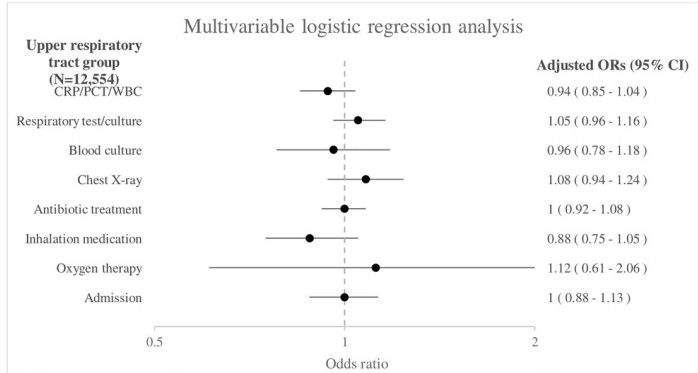

**Fig 2. Association between sex and management in the upper respiratory tract group (N = 12,554).** Boys as reference group. Adjusted for age, triage urgency, ill appearance, tachypnea, tachycardia, hypoxia, work of breathing, duration of fever, ED.

collected with detailed information on clinical presentation and management. Additionally, we performed subgroup analyses to describe the observed sex differences in the total group more in depth. However, this study has some limitations which should be mentioned. First, data on respiratory symptoms related to the upper respiratory tract such as runny nose and coughing were collected but data on findings of physical examination such as wheezing or focal abnormalities on auscultation were not collected, which is important since wheezing is often associated with a viral infection and clinicians may tend to start inhalation medication in these children [25]. Second, we did not take into account gender, which might play a role in how girls and boys present themselves, clinician's perception and therefore management. Gender refers to a person's self-presentation as male or female and the influence of the social and cultural environment [3]. However, we have performed a subgroup analysis in children up to five years to reduce behavioral factors associated with gender. Third, we did not adjust for other determinants such as parental concern, clinicians' experience or gut feeling and sociocultural factors. However, we have adjusted our analysis for important clinical factors related to disease severity including tachypnea and increased work of breathing. This is a first step in providing insight in the role of sex in management in febrile children with respiratory

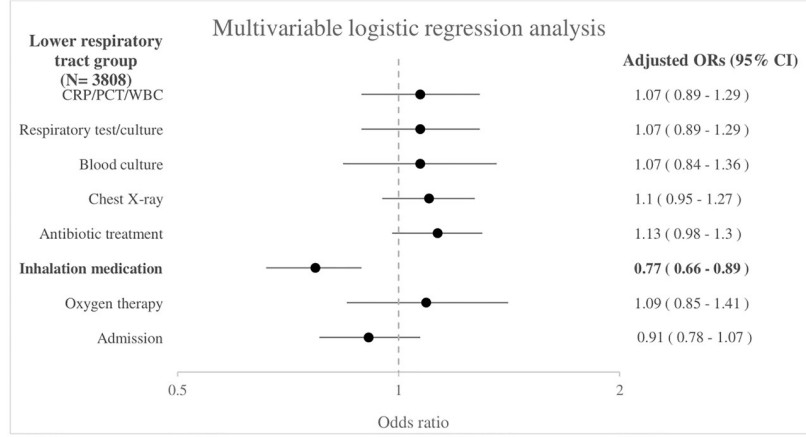

**Fig 3. Association between sex and management in the lower respiratory tract group (N = 3808).** Boys as reference group. Adjusted for age, triage urgency, ill appearance, tachypnea, tachycardia, hypoxia, work of breathing, duration of fever, ED.

symptoms in emergency medicine. Future research should focus on to what extend the observed differences between boys and girls are related to clinicians' attitudes or to differences in clinical symptoms at the time of presentation and disease severity.

## Conclusion

In a large cohort of previously healthy febrile children with respiratory symptoms attending European EDs, girls present less frequently with tachypnea and increased work of breathing compared with boys. Girls receive inhalation medication less frequently than boys, especially when they have a lower respiratory tract infection and are below five years of age. However, no sex differences were found regarding diagnostic tests, oxygen therapy and admission. Insight in these sex differences may lead to better understanding of the role of sex in pediatric emergency medicine.

## Supporting information

**S1 Fig. Phenotyping algorithm cause of infection.** *Patients could have a identified viral co-infection.
(PDF)

**S2 Fig. Association between sex and management in children below five years (N = 14,967).** Boys as reference group. Adjusted for age, triage urgency, ill appearance, tachypnea, tachycardia, hypoxia, work of breathing, duration of fever, ED.
(PDF)

**S1 Table. Management stratified by sex with range per ED (N = 19,781).** Boys as reference group. Absolute numbers and percentages (%) are shown.
(PDF)

**S2 Table. Association between sex and management (N = 19,781).** Boys as reference group. Adjusted for age, triage urgency, ill appearance, tachypnea, tachycardia, hypoxia, work of breathing, duration of fever, ED.
(PDF)

**S3 Table. Association between sex and management in children with an upper respiratory tract infection (N = 12,554).** Boys as reference group. Adjusted for age, triage urgency, ill appearance, tachypnea, tachycardia, hypoxia, work of breathing, duration of fever, ED.
(PDF)

**S4 Table. Association between sex and management in children with a lower respiratory tract infection (N = 3808).** Boys as reference group. Adjusted for age, triage urgency, ill appearance, tachypnea, tachycardia, hypoxia, work of breathing, duration of fever, ED.
(PDF)

**S5 Table. Subgroup analysis in children below five years (N = 14,967).** Boys as reference group. Adjusted for age, triage urgency, ill appearance, tachypnea, tachycardia, hypoxia, work of breathing, duration of fever, ED.
(PDF)

**S1 File. PERFORM consortium authors list.**
(PDF)

**S1 Checklist.**
(PDF)

## Author Contributions

**Conceptualization:** Chantal D. Tan, Soufiane el Ouasghiri, Ulrich von Both, Enitan D. Carrol, Marieke Emonts, Michiel van der Flier, Ronald de Groot, Jethro Herberg, Benno Kohlmaier, Michael Levin, Emma Lim, Ian K. Maconochie, Federico Martinon-Torres, Ruud G. Nijman, Marko Pokorn, Irene Rivero-Calle, Maria Tsolia, Clementien L. Vermont, Werner Zenz, Dace Zavadska, Henriette A. Moll, Joany M. Zachariasse.

**Data curation:** Chantal D. Tan, Soufiane el Ouasghiri.

**Formal analysis:** Chantal D. Tan, Soufiane el Ouasghiri.

**Investigation:** Chantal D. Tan, Soufiane el Ouasghiri.

**Methodology:** Chantal D. Tan, Soufiane el Ouasghiri, Joany M. Zachariasse.

**Supervision:** Henriette A. Moll, Joany M. Zachariasse.

**Writing – original draft:** Chantal D. Tan, Soufiane el Ouasghiri.

**Writing – review & editing:** Chantal D. Tan, Soufiane el Ouasghiri, Ulrich von Both, Enitan D. Carrol, Marieke Emonts, Michiel van der Flier, Ronald de Groot, Jethro Herberg, Benno Kohlmaier, Michael Levin, Emma Lim, Ian K. Maconochie, Federico Martinon-Torres, Ruud G. Nijman, Marko Pokorn, Irene Rivero-Calle, Maria Tsolia, Clementien L. Vermont, Werner Zenz, Dace Zavadska, Henriette A. Moll, Joany M. Zachariasse.

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
