## [Decision Letter · Decision Letter 0]

19 May 2022

PONE-D-22-05605Sex differences in febrile children with respiratory symptoms attending European Emergency Departments: an observational multicenter studyPLOS ONE

Dear Dr. Zachariasse,

Thank you for submitting your manuscript to PLOS ONE. After careful consideration, we feel that it has merit but does not fully meet PLOS ONE’s publication criteria as it currently stands. Therefore, we invite you to submit a revised version of the manuscript that addresses the points raised during the review process.

We look forward to receiving your revised manuscript.

Kind regards,

Kenneth Michelson, MD MPH

Academic Editor

PLOS ONE

Journal Requirements:

Additional Editor Comments (if provided):

Dear Dr. Zachariasse,

Thank you for this submission. I look forward to your revisions. Please pay particular attention to the reviewers' comments, particularly the major themes identified by both: (1) what is the relevance of this study? How will it help us clinically or otherwise? (2) Change or justify your inclusion criteria. In particular, why exclude patients without fever?

Reviewers' comments:

Reviewer's Responses to Questions

**Comments to the Author**

1. Is the manuscript technically sound, and do the data support the conclusions?

Reviewer #1: Partly

Reviewer #2: Yes

2. Has the statistical analysis been performed appropriately and rigorously? 

Reviewer #1: Yes

Reviewer #2: Yes

3. Have the authors made all data underlying the findings in their manuscript fully available?

Reviewer #1: Yes

Reviewer #2: Yes

4. Is the manuscript presented in an intelligible fashion and written in standard English?

Reviewer #1: Yes

Reviewer #2: Yes

5. Review Comments to the Author

Reviewer #1: In their paper, Sex differences in febrile children with respiratory symptoms attending European Emergency Departments: an observational multicenter study, the study authors aim to assess sex differences in presentation and management of febrile children with respiratory symptoms across twelve different European emergency departments in a roughly one-year period of time. The data set is quite large and finds statistically significant differences in both work of breathing on arrival and interventional therapies, but has some underlying issues with the methodology that need to be addressed.

Major comments:

- Is there a reason that you didn’t just look at children with respiratory symptoms? Why combine fever and respiratory? I would put in a statement explaining this choice, given that there are many patients with URI/LRI who don’t also have fever.

- The exclusion criteria should be re-explored. The authors excluded anyone with a co-morbidity, including asthma, but kept in patients who presented with other non-respiratory infections or neurologic complaints, such as seizure, which would seem to alter ER management and muddy the picture more than the presence of an underlying comorbidity. I think you need to decide to look at all-comers and then do subanalyses or decide you want to look to only at otherwise healthy children who are presenting with URI/LRI without another major presenting complaint. One of the main findings was that girls get more antibiotics overall, but this disappears when looking only at URI/LRI and can likely be explained by the higher prevalence of UTI – though we don’t have this data (only that girls have more co-infections of some kind). If you are truly looking at management of respiratory disease and trying to get at underlying sex differences, these kids need to be pulled out of the data.

- I would clarify further the purpose of this study – it’s hard for me to piece together the clinical significance. The last sentence of the introduction says that finding differences in how these children present “may improve diagnosis and treatment” – but there is no further discussion of this and it’s hard to say how this data would do this. The last sentence of the discussion seems more reasonable – that there are differences in care and next step is to determine if these are based on clinical symptoms or physicians’ attitudes (ie, some type of implicit bias). I think I would stick with this theme, that this is purely descriptive, but more work needs to be done to figure out why there are differences.

Minor comments:

Abstract

-Line 83: I’d include p values (or ORs) with these statistics, particularly since the differences are so small.

Introduction

- Line 97: Small point, but I’d replace “inequality” with “inequity” – there are many reasons that different groups receive different treatments, but the goal should be to provide equitable care – which is what I think you are trying to get at here.

- Line 100: Would change “adults in emergency medicine” to “adults presenting to the emergency department”

- Line 107: “research on sex differences in children remains scarce” – I don’t think this is true. If you look at the reference list for the next paper you talk about, there are multiple studies there. Much of the research coming out of pediatric EM will contain demographic data and comment on any sex difference in presentation. You may be able to say that we need more research into differences in emergency management by sex – since I agree, there’s not as much out there about management in particular.

- Line 108: “The first study to assess the role of sex in pediatric emergency…” I would definitely talk more about this study – ie, what they found. I think it leads nice into your study and puts it into more context. They looked into a huge group of kids and found that boy present more frequently to the ED, get more inhalational medications, and girls get more labs – but this was all pooled data and didn’t look at a specific population. Here, you are attempting to only look at febrile kids with respiratory symptoms so you can really get to whether or not there are differences that exist here. This will also make your next sentences about sex different in asthma/bronchiolitis make more sense since they seem a bit disconnected currently.

- Line 123: “may improve diagnosis and treatment” – this doesn’t seem to be the goal of your paper overall based on the discussion and conclusions. I think you mean the role of sex in the management of children in the emergency department. There is no discussion in the paper at all around improving diagnosis or improving treatments since there is no information on outcomes (ie did they receive a correct diagnosis or was that treatment useful), only descriptive information on who presented and what was done.

Methods

- Line 150: “We excluded children with missing data on management and children with comorbidity.” Two things

1. I would move your source for what defines a comorbidity here. It looks like the source is referencing a paper that uses ICD-9 codes (though your study was done during the ICD-10 era, so I would use an updated source – looks like there have been two newer versions, PMID: 29496546). Were these codes only present at the index visit? Or if the patient ever had this code?

2. Why did you exclude these children? Seems like if these are large university hospitals you are going to have a lot of kids with comorbidities. Particularly in a study that is looking at the use of inhalational medications which should be mostly used in children who carry a diagnosis of asthma (or should). Is it possible to include them and then do a subanalysis of children who did not have comorbidities? Or look at all the data and then control for the number of kids with asthma? If not – I would at least include information on how many kids (what %) you excluded due to a comorbidity.

- Line 160: “Children could have had gastrointestinal (diarrhea or vomiting) or neurological (seizures or focal neurological signs or meningeal signs) symptoms in addition to their respiratory symptoms.” Were they allowed to have other symptoms as well? Or were they excluded if they had other symptoms except fever, respiratory and possibly GI/neuro. I would clarify what you mean here.

- Line 174: “We categorized diagnostic tests into simple and advance.” This classification seems a bit arbitrary – there are many kids with fever/respiratory who get a CXR and go home on antibiotics. I wouldn’t consider that more advanced than doing bloodwork (which is often reserved for kids who are actually sicker since it’s a more invasive intervention). I would get rid of this distinction and just split into “imaging,” “bloodwork,” and “respiratory test/culture”.

- Line 186: How was ill appearance defined? Triage nursing? Physicians note?

- Line 193: It’s unclear to me what “behavioral and psychological processes” means and has to do with fever/respiratory symptoms and ED management. Please clarify what you mean here.

- Line 194: Please change odds ratios to adjusted odds ratios

Results

- Line 205: Please include the p value for these numbers, particularly since they are so similar between groups. Or, even better, include the odds ratio with the confidence interval.

- Table 1:

I think you should include the actual p values here instead of just the asterisk that it’s <0.05. Again – I think having an odds ratio with a confidence interval would also provide much more information than a p value. I also can’t tell what is significant – eg focus of infection is labeled as significant, but I can’t tell if one sex is more likely to have an “other” focus of infection, which will likely affect management. Same with cause of infection.

- Table 2/Figure 1.

I would combine this with Figure 1. It looks strange to have this table without any statistics on the significance – you have to then go to Figure 1 to find the actual aORs – but Figure 1 doesn’t include all of the variables in Table 2… so I still don’t know what the aOR of getting a CXR was. It’s a pretty figure, but I think it makes more sense to just add this information to a fourth column in Table 2 if editor agrees.

I would control for presence of another source of infection other than respiratory. This is certainly going to change management. As you say – more females had UTIs and they are going to then get antibiotics that are not for their respiratory disease. Instead of just adding as a limitation, you can control for this, particularly since one of your main findings that barely reached significance was antibiotic use. You should also consider removing these patients (making it an exclusion criteria) since this will so strongly affect management (I think this is the better option).

As a side note – the number of children who are getting interventions seem really high to me. From your S1_Table – it looks like one of your EDs sends blood work on 93% of the patients it sees with fever/respiratory symptoms which seems outrageous to me, particularly since you include any rhinorrhea or sore throat as a respiratory symptom and most of these patients are low-acuity. Is this correct? It feels like it can’t possibly be the same patient population.

Discussion

- Line 264: “This is in line with a previous study in which girls received less inhalation medication compared with boys in children with respiratory symptoms attending the ED (pooled OR 0.79, 95% CI 0.73 to 0.86).” Please move your reference for these statistics to after this line.

- Line 270: I’m having some trouble with this argument – please clarify here. It seems like you are saying that boys are more likely to wheeze, which is why they get more inhalational therapies – but then say this can’t be because we excluded kids with asthma. I think the truth is that if you are getting bronchodilators and are over the age of 2, you are going to get a diagnosis of asthma (since having wheeze/cough responsive to bronchodilators is essentially the definition). The kids in this study who received bronchodilators probably fell into one of two categories – either they just hadn’t had an ICD-10 code for asthma put on their chart yet, or they were under 2 years old and wheezing, so had a tentative diagnosis of viral induced wheeze and may or may not later be diagnosed with asthma. Either way – both of these are more prevalent in males (as you point out), which likely explains why they are getting more inhalational therapy.

- Line 274: Agree with this argument for more antibiotics in girls – but this is why you should exclude these patients with other reasons to get antibiotics if you really want to looks at management of respiratory complaints.

Other Figures

- See comments in results for Tables 1&2 and Figure 1

- I don’t think you need supplemental tables 2-4 – they are the same as Figures 1-3 with the inclusion of the unadjusted odds ratio, which you don’t need to include – though will leave to editor.

- No comments on other figures/tables

Reviewer #2: General comments:

The authors have performed an observational study evaluating the association of sex with characteristics of children visiting Emergency Departments for febrile respiratory illness. Strengths of the study include the inclusion of multiple centers, large number of subjects, and depth of available data.

Overall, the expected implications and intended applications of the study findings are unclear. Were the authors intending the inform the incorporation of patient sex into clinical decision making or even the development of a clinical decision rule? Or were they intending to shed light in potential sex-based disparities in emergency care, particularly with respect to management? There are parts of the introduction that suggest either objective. The authors' proposed significance of this study could be stated more clearly in the introduction and discussion.

Abstract

General: The conclusions are largely a reiteration of the results and instead could be nbetter used to discuss the implications of the results.

70: "0-18 years" could be interpreted to be inclusive of 18-year-olds, whom the study presumably excluded based on the age groups in Table 1.

79: Could mention that subgroups analyses were performed by upper versus lower respiratory tract infection

83: See comment on 205.

Methods

144: See comment on 70.

146: Why were shortness of breath, tachypnea, or wheezing not included as respiratory symptoms? These symptoms would capture children with greater illness severity who would be of higher interest clinically to pediatric emergency physicians than those with only fever and sore throat.

151: Exclusion of children with comorbidity needs to be explained more. What level of comorbidity was excluded, complex illness or all chronic conditions including even mild intermittent asthma? If all children who have received a diagnosis of asthma are excluded, it should be noted that there is significant variability as to which children with prior respiratory illnesses have received a chronic diagnosis of asthma versus only acute diagnoses of "bronchiolitis," "wheeze," etc. that may confound the study results.

Given the high prevalence of asthma and other chronic conditions in among children, the exclusion of children with comorbidity significantly limits the applicability of these results to the overall ED population. If this is intentional, the authors could be more explicit in stating that their study focuses only on previously healthy children.

164: How were children with concurrent infections allocated? Given that the percentages in Table 1 add up to 100%, presumably only one focus of infection was assigned to each child, but if one were to present with influenza and urinary tract infection, both are clinically significant and worth accounting for. The categories perhaps should not be treated as mutually exclusive.

187: Unclear what is meant by ED setting.

Results

200: In addition to the study sample size, this sentence should include the total initial number of febrile children with respiratory symptoms and the number that were excluded due to missing data or presence of morbidity.

205: The statement that girls had lower frequency of tachypnea is repeated multiple times across the manuscript but is questionable. The difference by sex, while perhaps statistically significant with the large sample size, is too small to be considered clinically important.

Discussion

General: The entirety of the discussion is in one large paragraph and may be more easily readable in divided into multiple paragraphs by topic.

257-264: These lines are largely a reiteration of the results, which seems unnecessary and better off being cut.

272: The authors hypothesize that the higher antibiotic prescription rate for girls may be due to a higher rate of urinary tract infections. This hypothesis could have been tested with their data by including a third subgroup for stratification for children with a non-respiratory focus of infection.

6. PLOS authors have the option to publish the peer review history of their article (what does this mean?). If published, this will include your full peer review and any attached files.

Reviewer #1: **Yes: **Alexandra H Baker

Reviewer #2: No

---

## [Author Response · Author response to Decision Letter 0]

29 Jun 2022

Manuscript reference number: PONE-D-22-05605

Title: Sex differences in febrile children with respiratory symptoms attending European Emergency Departments: an observational multicenter study

Dear dr. Emily Chenette and reviewers,

Thank you for reviewing our paper titled: "Sex differences in febrile children with respiratory symptoms attending European Emergency Departments: an observational multicenter study". We are very pleased to hear that a revised version of our manuscript is considered for publication in PLOS ONE. We sincerely thank the reviewers for their extensive comments to improve the paper, and have revised our manuscript according to their recommendations. The comments and suggestions on how to clarify the methodology and implications of our study were especially valuable. We hope that the revisions made to these sections made the overall paper more clear, and made it reproducible and more understandable for first-time readers. We feel that we were able to address all comments in the revised version and that our paper has improved significantly. Our point to point response to the individual review comments can be found below.

All co-authors have read and agreed upon the current submitted version.

Yours Sincerely, on behalf of all co-authors,

Prof. dr. H.A. Moll and dr. J.M. Zachariasse

---

## [Editor Report · Decision Letter 1]

11 Jul 2022

Sex differences in febrile children with respiratory symptoms attending European Emergency Departments: an observational multicenter study

PONE-D-22-05605R1

Dear Dr. Zachariasse,

We’re pleased to inform you that your manuscript has been judged scientifically suitable for publication and will be formally accepted for publication once it meets all outstanding technical requirements.

Kind regards,

Kenneth A Michelson, MD MPH

Academic Editor

PLOS ONE

Additional Editor Comments (optional):

Thank you for addressing reviewer concerns. I share some of reviewer 1's concerns about specific exclusions but appreciate the overall reframing of the article and inclusion of the UTI sensitivity analysis, which eliminates a disparity in antibiotic use.
---

## [Editor Report · Acceptance letter]

26 Jul 2022

PONE-D-22-05605R1 

Sex differences in febrile children with respiratory symptoms attending European Emergency Departments: an observational multicenter study 

Dear Dr. Zachariasse:

I'm pleased to inform you that your manuscript has been deemed suitable for publication in PLOS ONE. Congratulations! Your manuscript is now with our production department. 

Kind regards, 

on behalf of

Dr. Kenneth A Michelson 

Academic Editor

PLOS ONE